# ON EVALUATING EXPLAINABILITY ALGORITHMS

## ABSTRACT

A plethora of methods attempting to explain predictions of black-box models have been proposed by the Explainable Artificial Intelligence (XAI) community. Yet, measuring the quality of the generated explanations is largely unexplored, making quantitative comparisons non-trivial. In this work, we propose a suite of multi-faceted metrics that enables us to objectively compare explainers based on the correctness, consistency, as well as the confidence of the generated explanations. These metrics are computationally inexpensive, do not require model-retraining and can be used across different data modalities. We evaluate them on common explainers such as Grad-CAM, SmoothGrad, LIME and Integrated Gradients. Our experiments show that the proposed metrics reflect qualitative observations reported in earlier works.

## 1 INTRODUCTION

Over the past few years, deep learning has made significant progress, outperforming the state-of-the-art in many tasks like image classification (Mahajan et al., 2018), semantic segmentation (Zhu et al., 2018), machine translation (Kalchbrenner et al., 2016) and even surpassing humans in the games of Chess and Go (Silver et al., 2016). As these models are deployed in more mission-critical systems, we notice that despite their incredible performance on standard metrics, they are fragile (Szegedy et al., 2013; Goodfellow et al., 2014) and can be easily fooled by small perturbations to the inputs (Engstrom et al., 2017). Further research has also exposed that these models are biased in undesirable ways exacerbating gender and racial biases (Howard et al., 2017; Escudé Font & Costa-Jussà, 2019). These issues have amplified the need for making these black-box models interpretable. Consequently, the XAI community has proposed a variety of algorithms that aim to explain predictions of these models (Ribeiro et al., 2016; 2018; Lundberg & Lee, 2017; Shrikumar et al., 2017; Smilkov et al., 2017; Selvaraju et al., 2016; Sundararajan et al., 2017).

With such an explosion of interpretability methods (hereon referred to as explainers), evaluating them has become non-trivial. This is due to the lack of a widely accepted metric to quantitatively compare them. There have been several attempts to propose such metrics. Unfortunately, they tend to suffer from major drawbacks like computational cost (Hooker et al., 2018), inability to be extended to non-image domains (Kindermans et al., 2017a), or simply focusing only one desirable attribute of a good explainer. (Yeh et al., 2019).

In this paper, we propose a suite of metrics that attempt to alleviate these drawbacks and can be applied across multiple data modalities. Unlike the vast majority of prior work, we not only consider the correctness of an explainer, but also the consistency and confidence of the generated explanations. We use these metrics to evaluate and compare widely used explainers such as LIME (Ribeiro et al., 2016), Grad-CAM (Selvaraju et al., 2016), SmoothGrad (Smilkov et al., 2017) and Integrated Gradients (Sundararajan et al., 2017) on an Inception-V3 (Szegedy et al., 2015) model pretrained on the ImageNet dataset (ILSVRC2012) (Deng et al., 2009), in an objective manner (i.e., without the need of a human-in-the-loop). Moreover, our proposed metrics are general and computationally inexpensive. Our main contributions are:

1. Identifying and formulating the properties of a good explainer.

2. Proposing a generic, computationally inexpensive suite of metrics to evaluate explainers.

3. Comparing common explainers and discussing pros and cons of each.

We find that while Grad-CAM seems to perform best overall, it does suffer from drawbacks not reported in prior works. On the other hand, LIME consistently underperforms in comparison to the other models.

## 2 RELATED WORKS

The field of XAI has become an active area of research (Doshi-Velez & Kim, 2017; Lipton, 2016) with significant efforts being made to explain AI models, either by generating local (Ribeiro et al., 2016; Shrikumar et al., 2017; Sundararajan et al., 2017; Selvaraju et al., 2016; Smilkov et al., 2017) or global (Lundberg & Lee, 2017; Ribeiro et al., 2018) explanations.

Simultaneously, there are growing research efforts into methods to formally evaluate and compare explainers (Mohseni et al., 2018; Gunning, 2019; Wolf, 2019; Gilpin et al., 2019). Notably, Murdoch et al. (2019) introduced a framework with three desiderata for evaluation, viz. predictive accuracy, descriptive accuracy and relevancy, with relevancy judged relative to a human. In contrast, Hall et al. (2019) compiled a set of desired characteristics around effectiveness, versatility, constraints (i.e., privacy, computation cost, information collection effort) and the type of generated explanations, which do not need human evaluation, and therefore are objective. However, they focus very little on aspects such as correctness. Recently, DeConvNet (Noh et al., 2015), Guided BackProp (Springenberg et al., 2015) and LRP (Bach et al., 2015) have been shown to not produce theoretically correct explanations of linear models (Kindermans et al., 2017b). As a result, two explanation techniques, PatternNet and PatternAttribution, that are theoretically sound for linear models were proposed. Other efforts focus on evaluating saliency methods (Kindermans et al., 2017a; Adebayo et al., 2018) and show that they are unreliable for tasks that are sensitive to either data or model. Samek et al. (2017) and its variations (Hooker et al., 2018; Fong & Vedaldi, 2017; Ancona et al., 2018) infer whether a feature attribution is correct by measuring performance degradation when highly attributed features are removed. For instance, Hooker et al. (2018) shows that commonly used interpretability methods are less accurate or are on-par with a random designation of feature importance, whereas ensemble approaches such as SmoothGrad (Smilkov et al., 2017) are superior.

Yang & Kim (2019) proposed three complementary metrics to evaluate explainers: model contrast score – comparing two models trained to consider opposite concepts as important, input dependence score – comparing one model with two inputs of different concepts, and input dependence rate – comparing one model with two functionally identical inputs. These metrics aim to specifically cover aspects of false-positives. Alvarez-Melis & Jaakkola (2018) define an alternative set of metrics, around explicitness – intelligibility of explanations, faithfulness – feature relevance, and stability – consistency of explanations for similar or neighboring samples. Finally, Yeh et al. (2019) define and evaluate fidelity of explanations, namely quantifying the degree to which an explanation captures how the underlying model itself changes in response to significant perturbations.

Similar to previous work, we focus on objective metrics to evaluate and compare explainers. However, we not only consider correctness, but also consistency and confidence (as defined next).

## 3 WHAT MAKES AN EXPLAINER GOOD?

### 3.1 PRELIMINARIES

In the following discussions, let $x \in \mathbb{R}^n$ be an arbitrary data point and $y$ be the corresponding ground truth label from the dataset $\mathcal{D} = \{(x_i, y_i), 1 \leq i \leq M\}$. Let $f$ be the classifier realized as a neural network parameterized by weights $\theta$. Let $\mathcal{T}$ be the set of transformations under which semantic information in the input remains unchanged. If $t$ is an arbitrary transform from this set, let $t^{-1}$ be its inverse transform. For example, if $t = Rot-90°$, then $t^{-1} = Rot + 90°$. In general, $t^{-1}(t(x)) = x, \forall x \in \mathcal{D}$.

Let $E_f$ be any explainer that generates explanations for predictions made by classifier $f^1$.

---

[1] the subscript $_f$ is used only when its absence causes ambiguity.

Finally, let $d(,)$ be a function that computes the difference between the generated explanations[2]. For example, if the explainer $E$ generates saliency maps (e.g. GradCAM and SmoothGrad), $d$ could be a simple $\ell_p$ norm.

Additionally, in order to ensure that we minimize the impact that pathologies of the underlying classifier have on the properties we are interested in, we assume that the classifier has acceptable test-set performance. Furthermore, we also assume that the classifier performance does not degrade significantly under the transformations we perform (described in Sec. 3.2.2). If the classifier does not satisfy these conditions, it is prudent to improve its performance to acceptable levels prior to attempting to explain its outputs. One cannot extract reliable explanations from underperforming underlying models (Ghorbani et al., 2019; Samek et al., 2017).

## 3.2 DESIDERATA FOR A GOOD EXPLAINER

Inspired by earlier works on important aspects of an explainer's quality (Yang & Kim, 2019; Alvarez-Melis & Jaakkola, 2018; Yeh et al., 2019), our proposed evaluation framework consists of the following components:

- Correctness
- Consistency
- Confidence

We elaborate on these components as well as methods to compute them in the image classification scenario. Even though these are evaluated independently, they can be combined together to give a single scalar value to compare explainers in a straightforward way. However, the weight for each component depends heavily on the use case and end-user preference. This is beyond the scope of the current work and thus is not discussed further. Further, since we elaborate on the image classification scenario, we use inputs and images interchangeably with the understanding that the described methods or equivalents can be trivially adapted in other modalities.

### 3.2.1 CORRECTNESS

Correctness (sensitivity or fidelity in literature) refers to the ability of an explainer to correctly identify components of the input that contribute most to the prediction of the classifier. Most metrics proposed so far focus solely on correctness and attempt to compute it in different ways, often requiring retraining of the underlying classifier. Moreover, they do not capture all aspects of correctness nor do they generalize to other data modalities.

We propose a novel computationally-inexpensive method that addresses these drawbacks. It takes into consideration both that the explainer identifies most of the relevant components and does not incorrectly select non-important components as important.

If the explainers are performing as expected, a simple masking of the input image with the associated explanation should provide better accuracy as the network is unlikely to be confused by the non-important pixels. However, we do not observe this in practice, as we show empirically that vanilla masking results in severe performance deterioration (see Table 9 and 10 for results). We hypothesize that this is because of the following reasons:

- The masked image has a large proportion of empty pixels[3] and thus does not belong to the data distribution ($p_{data}$)
- Extracted pixels are important in the context of the background pixels, and as such removing context makes the masking meaningless. Additionally, Convolutions have the inductive bias that the neighbouring pixels are highly correlated that helps perform well on visual tasks (LeCun et al., 1998). Simple masking breaks this correlation.

Based on the above observations, we conclude that it is crucial to have a realistic background for the extracted patches to properly evaluate them. We propose the following procedure to provide a background such that the resulting image is closer the data distribution

---

[2]We do not require $d(,)$ to be a distance metric in the strictest sense.

[3]using the first convolution layer bias as a values for the blank pixels does not help either

For each class in the dataset, we select the top $k$ and bottom $k$ images, sorted in decreasing order based on the probability assigned by the classifier to the ground-truth class. We then randomly pair each of the top images with one of the bottom images. For each pair, we extract important regions identified by the explainer from the top image (i.e high confidence images) and overlap them over the corresponding bottom image (i.e low confidence images). We use the bottom $k$ images for this task as we know that they are uninformative for the classifier as evidenced by the assigned probability. We thus obtain a dataset of masked images with important regions from the most important images along with relevant yet non-informative backgrounds for each class (see Fig. 1 for an example). Formally, the masking operation can be represented as:

$$M = Th(E_f(H)) \otimes H + Th\left(1 - E_f(H)\right)) \otimes L \tag{1}$$

Where $M$ is the new masked image, a threshold function, $H$ the high confidence image, $L$ the low confidence image and $\otimes the element-wise multiplication operator.$

We then measure the accuracy on this masked dataset and compare it with the accuracy on the bottom $k$ images subset.

Note that the above mentioned process only evaluates if the explainer is capturing important pixels. In order to verify that the explainer does not select non-important pixels, we repeat the same process but instead use the inverted saliency map [4] and recompute accuracy on this dataset. In this scenario, we expect the accuracy to deteriorate.

Formally, the inverse masking process can be defined as follow:

$$M = Th\left(1 - E_f(H))\right) \otimes H + Th(E_f(H)) \otimes L \tag{2}$$

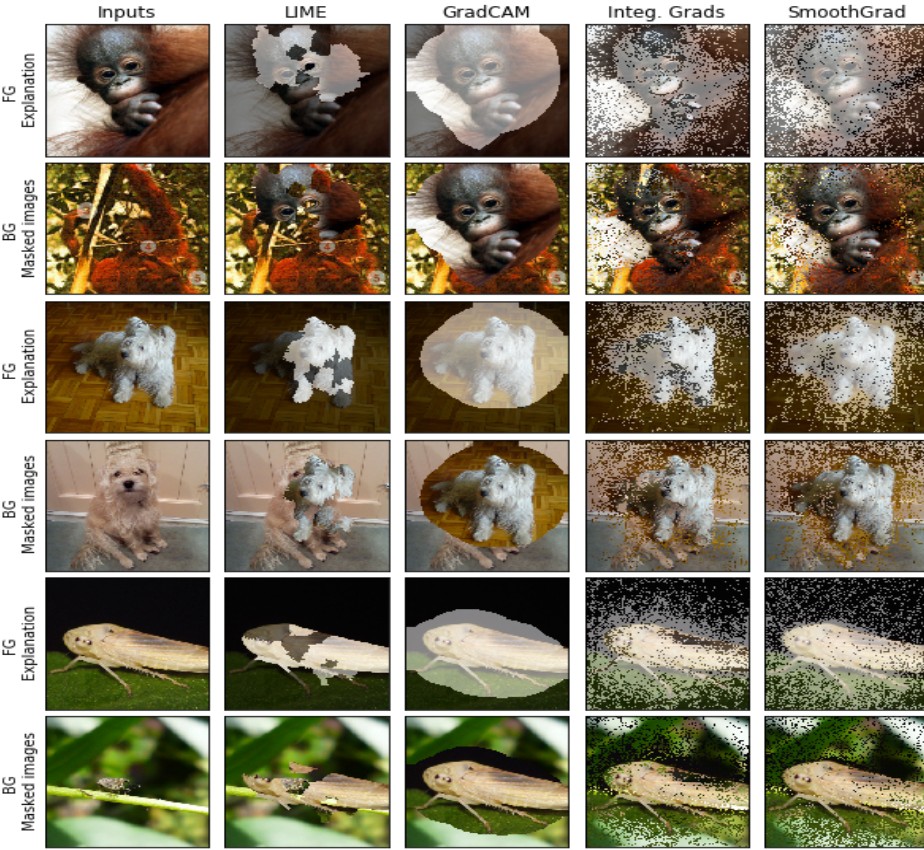

Figure 1: Examples of the proposed algorithm for correctness

Interestingly, these masked accuracies are similar to the precision and recall metrics used in information retrieval (Manning et al., 2009). This provides motivation to combine these differences into

---

[4]1. - saliency map as we pre-process the saliency map values to lie in $[0, 1]$ using min-max scaling

a *Pseudo-F1* score by computing the harmonic mean of accuracy on normal masked images and 1 - accuracy on inverse masked images. Formally this can be computed as:

$$Pseudo - F1 = \frac{2}{(1 - acc@1_{inverse})^{-1} + (acc@1)^{-1}} \qquad (3)$$

### 3.2.2 CONSISTENCY

We define consistency as the ability of the explainer to capture the same relevant components under various transformations to the input. More specifically, if the classifier predicts the same class for both the original and transformed inputs. Then, consistency measures whether the generated explanation for the transformed input (after applying an inverse transform) is similar to the one generated for the original input.

For example, if we apply vertical flip as a semantically invariant transformation, we flip the generated heatmap from the transformed image before comparing with the heatmap generated for the original image.

Formally, this can be represented as

$$d(E(f(x)), t^{-1}(E(f(t(x))))) \leq \epsilon \ \forall x \ s.t \ f(x) = f(t(x)), \ t \in \mathcal{T} \qquad (4)$$

**Semantically Invariant Transforms**   We focus on a subset of all potential transformations which does not change the semantic information contained in the input. We call this subset *Semantically Invariant Transforms*. Most work so far has considered only noising as a method of transforming the input. By constraining the magnitude of the added noise, we can control the size of the neighbourhood in which we perturb the images. In this work, we consider not only simple transformations that perturb the image in a small neighbourhood but also those that move the data point to vastly different regions in the input space while still retaining the semantic information contained within. This allows us to verify whether the explainer works as expected across larger regions of the input space.

For example, in the case of images, the family of transformations $T$ include affine transformations (translations and rotations), horizontal and vertical flips, noising (white, blue etc.), scaling etc.

In the image domain, $d$ could be realized as the $\ell_2$ (Euclidean) distance between explanations of the ground truth and inverted the transformed images (according to Eq. 4). However, Wang et al. (2005) and Zhao & Itti (2016) have shown that $\ell_2$ is not robust for images and may result in larger distances between the pairs of mostly similar explanations. This is attributed to the fact that $\ell_2$ is only a summation of the pixel-wise intensity differences and, as a result, small deformations may results in large distances. Even when the images are normalized before hand, $\ell_2$ is still not a suitable distance for images.

Therefore, we instead use Dynamic Time Warping (DTW) (Sakoe & Chiba, 1978) which allows for computing distances between two time series, even when misaligned or out of phase. Ibrahim & Valli (2008) has shown that DTW is effective for images as well, not only for temporal data as originally intended. Due to DTW's high computational cost (quadratic time and space complexity), we use FastDTW (Salvador & Chan, 2007), an approximation of DTW that has linear complexity in order to compute the distance between pairs of explanations.

### 3.2.3 CONFIDENCE

Finally, confidence is concerned with whether the generated explanation and the masked input result in high confidence predictions. This is a desirable property to enable explanations to be useful for downstream processes including human inspection (Hall et al., 2019). So far, our method for computing correctness sheds light only on the average case and is not particularly useful for individual explanations.

Generating high-confidence predictions is related to the well researched field of max-margin classifiers (Gong & Xu, 2007). A large margin in classifiers is widely accepted as a desirable property. Here, we extend this notion to explainers and propose that explainers generating explanations that result in high confidence predictions are desirable to those that do not. In addition to the desirable statistical properties that this enforces, high confidence predictions are also vital for building trust

with human users of the explainer as they are more interested in the per-instance performance than the average (Narayanan et al., 2018; Ross et al., 2017).

Concretely, we use the same procedure as in Sec. 3.2.1. Instead of computing the increase in accuracy, we compute instead the difference in probability assigned to the ground-truth class, as well as the difference in entropy of the softmax distributions of the original and masked images. We report this for both normal and inverted saliency maps. We expect to observe a positive probability difference and negative entropy difference under normal masking and an inverted behavior under inverse masking owing to similar reasons discussed in Sec. 3.2.1.

However, explainers that generate coarse explanations can easily fool this metric. An extreme case is when the explainer considers the entire input as useful. Such an explainer is useless but will have the theoretically highest change in confidence and entropy. To combat this and to establish how sparse the generated explanations are, we also report the average number of pixels in the explanations, normalized by the total number of pixels in the image.

We do not combine these numbers into one as different situations have different preferences. For example, in computational biology domains, sparsity is not as important as increase in confidence. The right weighting again depends on the use case and user preference.

## 4 EXPERIMENTS

We use an Inception v3 (Szegedy et al., 2015) architecture pretrained on ImageNet (ISLVC-2012) [5]. We compare LIME, Grad-CAM, SmoothGrad and Integrated Gradients and measure how they perform on the metrics described previously. All experiments and explainers (except LIME) were implemented in PyTorch (Paszke et al., 2017). Wherever possible, we reused the official implementation or kept our re-implementation as close to the official codebase as possible. The correctness and confidence metrics for every explainer are computed over 5 runs and mean values are reported. We use a fixed constant threshold to binarize explanation masks. We conducted further experiments by thresholding on the percentiles [6] instead(as done in (Smilkov et al., 2017)). These results have been reported in tables 6 and 7. We found that the choice did not affect the relative trends observed.

We consider the following semantically invariant transforms: translations ($x = \pm 0.2$, $y = \pm 0.2$), rotations ($-15°$, $-10°$, $-5°$, $5°$, $10°$, $15°$), flips (horizontal and vertical). To establish that these do not produce too many out-of-distribution samples (causing a decrease in classifier performance), we compute the accuracy of the underlying classifier under these transformations. Table 1 shows that, indeed, drops in accuracy are not significant.

Even though noising is semantically invariant in the image domain, we do not consider it in our experiments as some explainers like Smoothgrad would be unfairly favoured.

Table 1: Baseline accuracies under transformations

| Transformation | Param | Acc@1 | Acc@5 |
|---|---|---|---|
| No Transformation | | 77.21 | 93.53 |
| Horizontal Flip | | 77.19 | 93.45 |
| Vertical Flip | | 50.72 | 75.33 |
| Translation | x = 0.2 | 75.46 | 92.41 |
| | x = -0.2 | 75.21 | 92.40 |
| | y = 0.2 | 74.90 | 92.00 |
| | y = -0.2 | 74.87 | 92.04 |
| Rotation | $-15°$ | 68.29 | 87.49 |
| | $-10°$ | 71.19 | 89.35 |
| | $-5°$ | 74.62 | 91.90 |
| | $5°$ | 74.76 | 92.08 |
| | $10°$ | 71.35 | 89.40 |
| | $15°$ | 67.99 | 87.18 |

---

[5]The weights were taken from pytorch-hub
[6]Percentiles are computed on individual explanation map pixels

Table 2: Accuracy on the bottom 5 images after masking, using normal or inverse masking, when normal accuracy of the bottom 5 images with no masking is acc@1=11.42 and acc@5=53.8

|  | LIME | Grad-CAM | Integ. Grad. | Smoothgrad |
|---|---|---|---|---|
| acc@1 | 55.03 | 97.44 | 61.6 | 76.22 |
| acc@1 inverse | 42.89 | 15.4 | 10.64 | 9.74 |
| Pseudo-F1 | 0.56 | 0.91 | 0.73 | 0.83 |
| acc@5 | 80.34 | 99.59 | 80.72 | 89.72 |
| acc@5 inverse | 70.08 | 51.4 | 28.42 | 30.95 |
| Pseudo-F1 | 0.44 | 0.65 | 0.76 | 0.78 |

## 4.1 EVALUATING CORRECTNESS

We perform the procedure described in Sec. 3.2.1 and report the results in Table 2. The baseline acc@1 and acc@5 were 11.42% and 53.8% respectively. We hypothesized that a good explainer's accuracy increases with the normal masking and decreases with inverse masking. We see the expected increases in accuracies across all the explainers with Grad-CAM obtaining the highest increase at 97.44%.

However, for the inverse masking, we see that both LIME and Grad-CAM show results contrary to our hypothesis. This can be explained by observing examples of maps generated in Figs. 1 and 4. We see that, on average, Grad-CAM generates much larger explanations than all other explainers (can be seen in Table 3 as well). This implies that Grad-CAM misidentifies several non-important pixels as important and thus when we compute the inverse masks, we remove non-important pixels that could confuse the classifier.

In the case of LIME, we again see from Table 3 and Figs. 1 and 4 that LIME generates the smallest explanations. We further see from Table 2 that LIME has the smallest accuracy gain (in both acc@1 and acc@5). These indicate that LIME fails to select important pixels that were selected by all other explainers. Thus, we can conclude that the inverse masks in case of LIME would contain important pixels as well and thus would cause increase in accuracy as observed.

### 4.1.1 EFFECT OF NUMBER OF IMAGES

As detailed previously, our methodology for computing correctness involves choosing a number $k$ of top and bottom images to be used for masking. We evaluate how sensitive the measured correctness of explainers are to the value of $k$. We report the changes in accuracy with respect to the unmasked bottom images for $k=\{5,10,15,20,25\}$ in Fig. 2. The actual accuracy numbers are also reported in Tables 4 and 5.

We see that for both acc@1 and acc@5, the change in accuracy for normal masking decreases as we increase $k$. This is as expected since the average confidence gap between the top-k and the bottom-k images decreases as $k$ increases. This means that the important pixels in the background images are masked with non-important pixels from the foreground images. On the contrary, LIME shows a smaller decrease in accuracy (both acc@1 and acc@5). This can be explained by the fact that LIME does not capture all important pixels, and therefore all important pixels from the background are not replaced by less-informative pixels.

Similarly, for acc@1 and acc@5 for inverse masking, we see that LIME, Smoothgrad and Integrated Gradients behave as expected, i.e., the drop in accuracy is diminished with $k$ is increased as we are retaining the informative parts from the new background images. Interestingly, the drop in accuracy for Grad-CAM is stable and close to zero. To understand this, we refer again to Table 3 and note that Grad-CAM produces the smallest inverse maps on average. This implies that when we perform the inverse masking, we retain much of the informative pixels of the background image and thus do not see significant drops in accuracy relative to the unmasked bottom-$k$ image dataset.

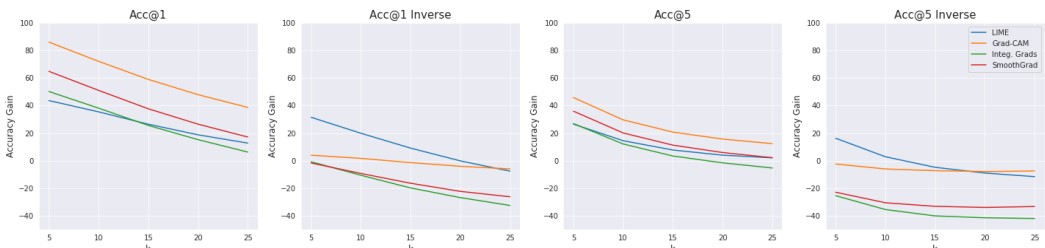

Figure 2: Effect of $k$ on correctness with normal and inverse masking

## 4.2 EVALUATING CONSISTENCY

Next, we evaluate consistency by computing the distance with FastDTW between the saliency maps generated on the original images and those generated when transformations are applied to the input image (Following Eq. 4). Fig. 3 and Table 8 report the normalized distances relative to each transformation (i.e., heatmaps sum to 1).

First, as transformations become more drastic relative to the original saliency maps, the distances also increase. This is the desired behavior one would expect, thus motivating our choice for using FastDTW. Second, Grad-CAM outperforms all other explainers, as reflected by the fact that its corresponding distances are always smallest. It is followed by Smoothgrad, Integrated Gradients and LIME. This is expected given the grainy saliency maps obtained with Integrated Gradients and Smoothgrad, as well as the patchy heatmaps generated with LIME.

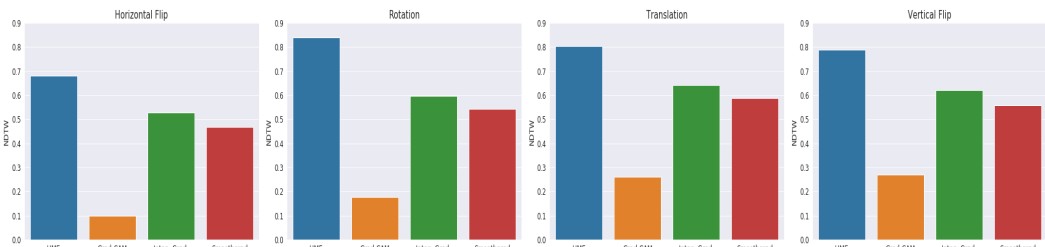

Figure 3: Normalized DTW distances across invariant transforms

## 4.3 EVALUATING CONFIDENCE

Measuring confidence quantifies the performance of the explainers on a per-instance case and not only in the average. As described in Sec. 3.2.3, we compute the change in probability assigned to the ground-truth class ($\Delta$ conf) as well as the change in entropy of the softmax distribution ($\Delta$ entropy) as proxies for estimating the confidence of explanations. Additionally, we report the proportions of pixels in the heatmaps to the total number of pixels[7], averaged across the top-$k$ dataset.

We see that for confidence, the trends mimic the ones observed in Table 2. This implies that masking with extracted heatmaps not only increases accuracy but also results in high-confidence predictions across explainers. More specifically, we see that Grad-CAM again outperforms the other explainers (both $\Delta$ conf and $\Delta$ entropy) in the normal heatmaps by large margins. In the case of inverse masking, confidence and entropy for LIME show behaviours contrary to our expectations. This can be attributed to the "patchiness" of the explanations generated by LIME which was discussed in the previous sections.

---

[7]89401 in the standard ImageNet preprocessing pipeline for Inception-v3

Table 3: Confidence of explanation (inverse refers to masking with the inverse saliency map)

|  | LIME | Grad-CAM | Integ. Grad. | Smoothgrad |
|---|---|---|---|---|
| $\Delta$ conf | 0.34 | 0.83 | 0.4 | 0.54 |
| $\Delta$ conf inverse | 0.23 | 0.03 | -0.01 | -0.01 |
| $\Delta$ entropy | -0.17 | -2.41 | -1.04 | -1.62 |
| $\Delta$ entropy inverse | -0.23 | 0.19 | 0.65 | 0.46 |
| avg. map size | 0.22 | 0.62 | 0.47 | 0.57 |

## 5 CONCLUSIONS AND FUTURE WORK

In this paper, we formulated desired properties of a good explainer and proposed a generic, computationally inexpensive suite of metrics – correctness, consistency and confidence – to objectively evaluate and compare explainers. We compared well-known explainers, such as LIME, Grad-CAM, Integrated Gradients and SmoothGrad, on a pretrained Inception-V3 model on the ImageNet dataset. Our experiments show that the metrics proposed capture various pros and cons of each explainer allowing users to make an informed choice about which explainer to use for their use case.

Specifically, we observe that Grad-CAM often performs better than the other explainers but suffers from drawbacks when inverse masking situations are considered. On the other hand, LIME performs poorly in all situations we consider.

Moreover, we also point out the pitfalls of trying to combine results from multiple metrics as they tend to hide anomalous behaviours of the underlying metrics (as seen from Pseudo-F1 from Table 2). We recommend that users sanity-check explainers by looking at individual metrics before making a decision based on the combined metric.

Furthermore, we urge the XAI community to resist the temptation to propose an one-size-fits-all metrics as we have shown that such metrics tend to hide nuanced trade-offs that practitioners need to be aware of.

Going forward, we invite the research community to test our metrics on other explainers, datasets, underlying classifiers and data modalities. Additionally, since the metrics proposed are differentiable, we believe exciting new liens of research would be to develop explainers that directly optimize for these metrics, as well as self-explaining models that incorporate such metrics into their learning regiment.

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

# A ADDITIONAL EXPERIMENTAL RESULTS

## A.1 VISUAL MASKING RESULTS

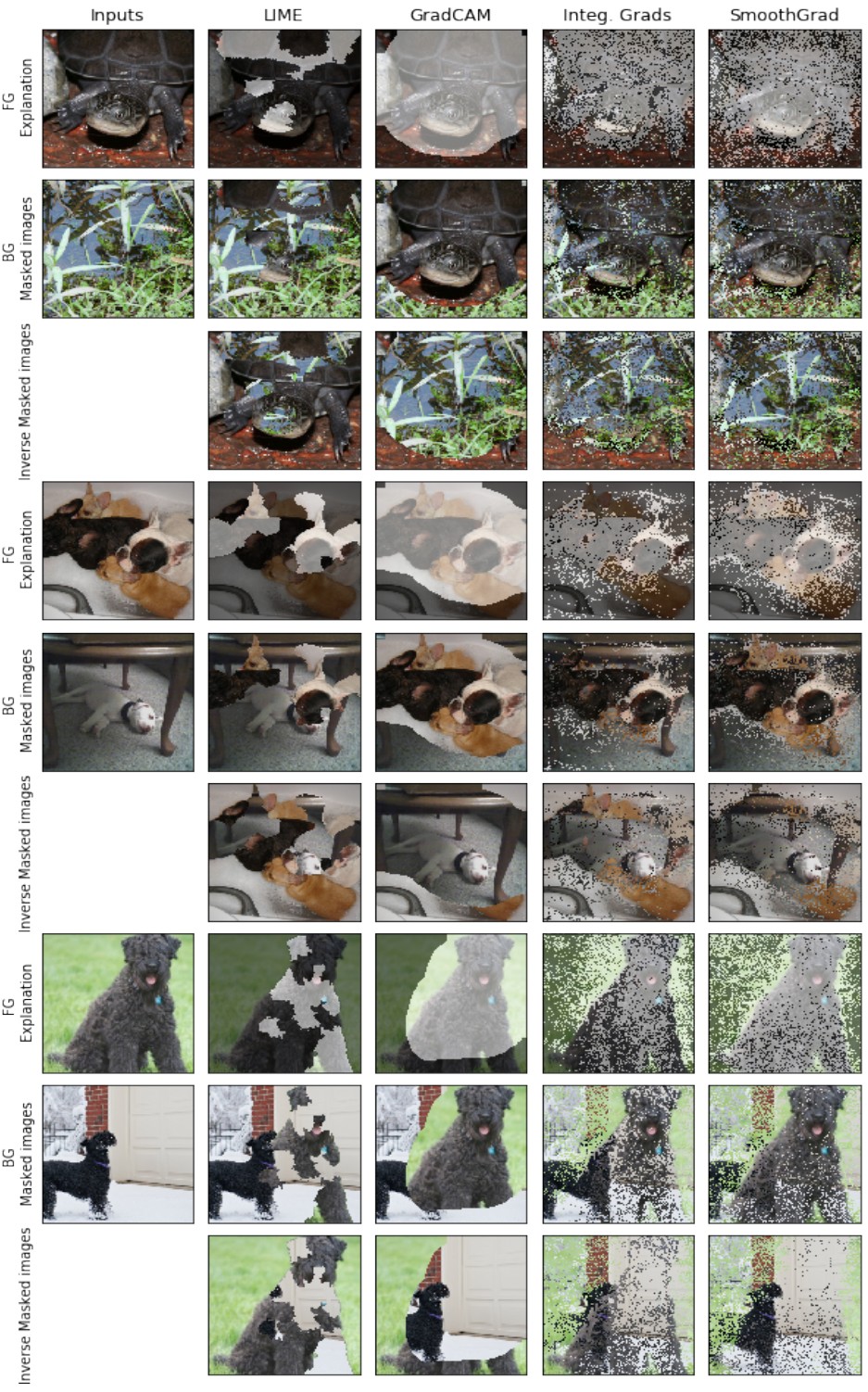

Figure 4: Examples of the proposed algorithm for correctness

## A.2 EFFECT OF $k$ ON CORRECTNESS FOR NORMAL AND INVERSE MASKING

Table 4: Effect of $k$ on correctness with normal masking

| $k$ | Original | | LIME | | Grad-CAM | | Integ. Grad. | | Smoothgrad | |
|---|---|---|---|---|---|---|---|---|---|---|
| | acc@1 | acc@5 | acc@1 | acc@5 | acc@1 | acc@5 | acc@1 | acc@5 | acc@1 | acc@5 |
| 5 | 11.42 | 53.8 | 55.03 | 80.34 | 97.44 | 99.59 | 61.6 | 80.72 | 76.22 | 89.72 |
| 10 | 25.39 | 70.01 | 60.8 | 84.5 | 97.44 | 99.63 | 63.5 | 82.14 | 76.39 | 90.13 |
| 15 | 37.91 | 78.88 | 64.37 | 86.64 | 96.87 | 99.63 | 63.49 | 82.3 | 75.55 | 90.19 |
| 20 | 48.18 | 83.9 | 66.95 | 87.92 | 96.08 | 99.59 | 63.44 | 82.35 | 74.68 | 89.83 |
| 25 | 56.23 | 87.06 | 69.0 | 89.15 | 94.86 | 99.44 | 62.5 | 81.77 | 73.4 | 89.22 |

Table 5: Effect of $k$ on correctness with inverse masking

| $k$ | Original | | LIME | | Grad-CAM | | Integ. Grad. | | Smoothgrad | |
|---|---|---|---|---|---|---|---|---|---|---|
| | acc@1 | acc@5 | acc@1 | acc@5 | acc@1 | acc@5 | acc@1 | acc@5 | acc@1 | acc@5 |
| 5 | 11.42 | 53.8 | 42.89 | 70.08 | 15.4 | 51.4 | 10.64 | 28.42 | 9.74 | 30.95 |
| 10 | 25.39 | 70.01 | 45.29 | 72.87 | 27.09 | 64.02 | 14.83 | 34.59 | 16.14 | 39.51 |
| 15 | 37.91 | 78.88 | 47.04 | 74.1 | 36.52 | 71.63 | 18.16 | 38.85 | 21.56 | 45.8 |
| 20 | 48.18 | 83.9 | 48.05 | 74.95 | 44.14 | 76.06 | 21.42 | 42.52 | 25.94 | 50.01 |
| 25 | 56.23 | 87.06 | 48.64 | 75.51 | 50.26 | 79.63 | 23.78 | 45.17 | 30.06 | 53.81 |

## A.3 EFFECT OF GRADIENT MAP THRESHOLD VALUE ON CORRECTNESS AND CONFIDENCE

Table 6: Accuracy on the top/bottom 5 images after percentile threshold masking, using normal or inverse masking, when normal accuracy of the bottom 5 images with no masking is acc@1=11.42 and acc@5=53.8

| Threshold Percentile | | acc@1 | acc@1 inverse | acc@5 | acc@5 inverse |
|---|---|---|---|---|---|
| 20th Percentile | Grad-CAM | 99.48 | 14.00 | 99.92 | 52.89 |
| | Integrated Gradient | 91.15 | 12.37 | 96.928 | 41.29 |
| | Smoothgrad | 91.51 | 11.95 | 97.17 | 41.0 |
| 40th Percentile | Grad-CAM | 96.81 | 17.39 | 99.38 | 53.93 |
| | Integrated Gradient | 75.45 | 11.12 | 88.02 | 33.13 |
| | Smoothgrad | 75.18 | 10.73 | 88.12 | 32.90 |
| 60th Percentile | Grad-CAM | 88.92 | 25.91 | 97.06 | 59.66 |
| | Integrated Gradient | 52.00 | 12.91 | 70.55 | 30.94 |
| | Smoothgrad | 49.38 | 11.7 | 68.43 | 29.72 |
| 80th Percentile | Grad-CAM | 63.85 | 49.62 | 87.3 | 76.72 |
| | Integrated Gradient | 25.10 | 27.79 | 48.56 | 49.17 |
| | Smoothgrad | 23.68 | 26.37 | 46.65 | 47.73 |

Table 7: Confidence of explanation using different thresholds

| Threshold | | Δ conf | Δ conf inverse | Δ entropy | Δ entropy inverse | avg. map size |
|---|---|---|---|---|---|---|
| 20th percentile | Grad-CAM | 0.869 | 0.018 | -2.69 | 0.13 | 0.8 |
| | Integ. Grad. | 0.737 | 0.007 | -2.36 | 0.28 | 0.8 |
| | Smoothgrad | 0.738 | 0.005 | -2.30 | 0.27 | 0.8 |
| 40th percentile | Grad-CAM | 0.816 | 0.043 | -2.46 | 0.21 | 0.6 |
| | Integ. Grad. | 0.5533 | -0.004 | -1.76 | 0.41 | 0.6 |
| | Smoothgrad | 0.5488 | -0.005 | -1.81 | 0.39 | 0.6 |
| 60th percentile | Grad-CAM | 0.693 | 0.102 | -1.67 | 0.23 | 0.4 |
| | Integ. Grad. | 0.321 | 0.004 | -0.72 | 0.54 | 0.4 |
| | Smoothgrad | 0.304 | -0.003 | -0.74 | 0.46 | 0.4 |
| 80th percentile | Grad-CAM | 0.42 | 0.281 | -0.42 | -0.20 | 0.2 |
| | Integ. Grad. | 0.09 | 0.111 | 0.39 | -0.11 | 0.2 |
| | Smoothgrad | 0.088 | 0.103 | 0.45 | -0.05 | 0.2 |

## A.4 NORMALIZED DTW ON SALIENCY MAPS UNDER INVARIANT TRANSFORMS

Table 8: Normalized DTW computed on the generated saliency maps under transformations

| Parameter | Parameter | LIME | Grad-CAM | Integ. Grad. | Smoothgrad |
|---|---|---|---|---|---|
| Horizontal Flip | | 0.67981 | 0.1006316 | 0.52763 | 0.468309 |
| Vertical Flip | | 0.7894171 | 0.2697139 | 0.62144 | 0.558899 |
| Translation | x = 0.2 | 0.789417 | 0.2428162 | 0.60073 | 0.546869 |
| | x = -0.2 | 0.793548 | 0.2452753 | 0.60145 | 0.54723 |
| | y = 0.2 | 0.812584 | 0.2688328 | 0.68498 | 0.633491 |
| | y = -0.2 | 0.81300 | 0.2847767 | 0.6845 | 0.622429 |
| Rotation | −15° | 0.8662869 | 0.2046104 | 0.61906 | 0.564211 |
| | −10° | 0.844156 | 0.1830615 | 0.60026 | 0.545514 |
| | −5° | 0.7984 | 0.1449023 | 0.56954 | 0.515016 |
| | 5° | 0.806391 | 0.1452136 | 0.56956 | 0.514659 |
| | 10° | 0.84784 | 0.1827558 | 0.60129 | 0.545972 |
| | 15° | 0.86896 | 0.2038526 | 0.61984 | 0.564713 |

## A.5 EFFECT OF USING ZERO AND GREY MASKING FOR CORRECTNESS

Table 9: acc@1 and acc@5 obtained after using zero masking

| | Original | LIME | Grad-CAM | Integ. Grad. | Smoothgrad |
|---|---|---|---|---|---|
| acc@1 | 77.21 | 48.3 | 72.64 | 25.03 | 28.62 |
| acc@5 | 93.53 | 66.86 | 89.69 | 42.95 | 49.03 |

Table 10: acc@1 and acc@5 obtained after using grey masking (i.e multiplying images with gradient maps and filling the background with the bias of the first convolutional filter in the model)

| | Original | LIME | Grad-CAM | Integ. Grad. | Smoothgrad |
|---|---|---|---|---|---|
| acc@1 | 77.21 | 48.36 | 72.54 | 24.86 | 28.53 |
| acc@5 | 93.53 | 66.88 | 89.59 | 42.72 | 48.81 |

