# OpenReview forum: "On Evaluating Explainability Algorithms"
_ICLR.cc/2020/Conference — Reject_

### Official Review · AnonReviewer3 · 2019-10-18
**Official Blind Review #3**

**Rating:** 1

**Review:**

This paper studies the interesting question of comparing the deep network visualization algorithms quantitatively. Several metrics are proposed, including correctness, consistency and confidence.

I like the notion of consistency, where an explainer should produce the same explanation under transformations of the image that does not change its “semantic content”.

However, I am confused or unconvinced by several arguments made in the paper, and if the authors can clarify them I am willing to increase my review. I think the major issue is that most metrics are justified with flimsy arguments, not compared with prior work, and do not lead to consistent ranking of the models.

Correctness: I am not convinced by the correctness evaluation for several reasons

1. The combined image is still out of distribution, and it is unclear why this is better compared to e.g. using a white background.
2. Does it favor visualization methods with a blob-shaped saliency map vs. scattered dots shaped saliency map? Does it favor methods with a larger salient region? For example, just from visual appeal, I do not think smoothgrad is worse than gradCAM, but the number says otherwise. I think the arbitrariness of this metric makes the numbers hard to believe.
3. If the original image is already incorrectly classified (since they are the ones where the classifier assigns the lowest probability) it is hard to imagine that adding random background can make the performance worse.

Therefore, it is also unclear what to make of the numbers in e.g. Table 2. There are so many metrics, precision, recall, F1, and none of them seem particularly well justified. They also do not rank the model in the same way. Which result should a practitioner believe?

Confidence: I am not sure the confidence vs. number of pixels comparison are useful. Across all methods, it seems to be more pixels -> increased confidence, which is unsurprising. I think the results are only useful if one method pareto dominates another, which is not what is observed in the experiments.

I do not understand the difference between confidence and correctness. It seems like both measure how well a model can predict the correct class given only the salient region. For example, if method A has higher confidence and lower correctness compared to method B, what does that mean? Under which situation should one choose method A over method B?

**Experience Assessment:**

I have published one or two papers in this area.

**Review Assessment: Checking Correctness Of Derivations And Theory:**

I assessed the sensibility of the derivations and theory.

**Review Assessment: Checking Correctness Of Experiments:**

I assessed the sensibility of the experiments.

**Review Assessment: Thoroughness In Paper Reading:**

I read the paper at least twice and used my best judgement in assessing the paper.

---

> ### Author Response · Authors · 2019-11-15
> **Response to ReviewerIII - pt 1**
>
>
>
> We would like to thank the reviewer for their valuable comments. We address their concerns below.
>
> Not compared with prior work:
> Response:
> In our literature review, we have not encountered prior work that deals with multi-faceted evaluation of explainers without involving human-in-the-loop baselines. We would be glad to know from the reviewer of any relevant work that we might have missed in our review.
>
> Proposed metrics do not lead to consistent ranking of models (explainers):
> Response:
> If every metric gives the same ranking then there’s no need to evaluate across multiple dimensions (which is the purpose of our paper). The fact that we find inconsistent rankings across the different metrics exemplifies that there’s no obviously one best explainer. This further justifies evaluating across multiple dimensions. This also emphasizes that users must be more explicit in their requirements and choose the explainer that’s best suited for the task at hand (eg. medical diagnosis, self driving cars etc.) and not blindly follow suggestions from other use cases.
>
> The combined images are still out of distribution:
> Response:
> Kindly see responses to reviewer1 and reviewer2
>
> Why is our proposed masking method better than white / black background.
> Response:
> In our experiments, we saw that the white / back background do not result in explanations that behave as expected (See Tables 9 and 10 in the appendix). Additionally, we also refer the reviewer to our responses to Reviewer2 where we have addressed a similar concern in a more detailed manner.
>
> Does the evaluation framework favor visualization methods with a blob-shaped saliency map over scattered dots shaped saliency map?
> Response:
> In our experience, that seems to be the case. This is also because the underlying classifier, a CNN, also prefers the blobs due to its inductive biases. As a result, the  accuracy increase is observed more prominently for blob shaped explanations.
>
> Does it favor methods with larger salient regions?
> Response:
> No, because we also measure inverse masking. If the explainer naively captures a large region to “cheat” the correctness metric, it will be penalized because the inverse mask would be too small and we will not see expected changes in accuracy. Additionally, we also report average explanation size (Avg. proportion of pixels in the explanations wrt to the total number of pixels in the image) which further allows us to detect if the explainer is trying to fool the metric by producing large explanations.
>
> If the original image is already incorrectly classified (since they are the ones where the classifier assigns the lowest probability) it is hard to imagine that adding random background can make the performance worse.
> Response:
> We actually only use the incorrectly classified/low confidence image for defining a background and the explanation map on the latter is ignored. The foreground (i.e., the relevant features) is extracted from the highly confident image. Since the prediction confidence is high, a proper explainer should highlight relevant class pixels on this image.
>
> Metrics like Precision, Recall and F1 are not well justified.
> Response:
> These metrics are widely used in classification and ranking tasks. One can indeed view the task of generation of explanation as being analogous to the retrieval task (as noted in [1]). Following this formulation, we try to map the changes in accuracy we see with our normal masking and inverse masking experiments to these metrics which are well understood by the community. We do acknowledge that the mapping is not one-to-one and thus label our metrics with a “pseudo” prefix.
>
> References:
> [1] Samek et.al, Evaluating the visualization of what a Deep Neural Network has learned

---

> > ### Author Response · Authors · 2019-11-15
> > **Response to ReviewerIII - pt 2**
> >
> > Efficacy of confidence vs. number of pixels comparisons
> > Response:
> > It is true that confidence alone is not enough to judge the quality of an explainer as it favorizes low sparsity explanation maps (i.e a naive explainer which flags all pixels in the image will.have the highest score). This is the reason why we report the average size of the selected patch which is a proxy for measuring the precision of our metric. Hence the best explainer should select the fewest number of pixels in an image while keeping a high masking score and low inverse masking score.
> >
> > Difference between confidence and correctness:
> > Response:
> > While the two numbers do come from the same experiment, we would like to point out that they are complementary to each other. Correctness gives us a coarse, distribution level view, i.e, how well the explainer performs on average. Confidence on the other hand, also informs us about the per-instance behaviour. In some instances, especially in the medical diagnosis domains, per-instance behaviour is more important than a distribution-level statistic. Additionally,  if method A and B have similar correctness scores, Confidence and Entropy (reported in table. 3) gives us more in-depth information about the per-sample performance of these methods, allowing us to differentiate between the performance of both models
> >
> >
> > Thanks,
> > The Authors

---

### Official Review · AnonReviewer2 · 2019-10-22
**Official Blind Review #2**

**Rating:** 1

**Review:**

See post-rebuttal updates below!

Summary
---

(motivation)
There are lots of heat map/saliency/visual explanation approaches that try to deep image classifiers more interpretable.
It's hard to tell which ones are good, so we need better ways of evaluating explanations.
This paper proposes 3 such explanation evaluation metrics, correctness, consistency, and confidence.

(approach - correctness)
An explanation is correct if it highlights enough of an image for a classifier to tell the correct class with only the highlight parts of the image.
The default way to evaluate on only highlighted portions is to set the non-highlighted bckground to black/grey.
Instead, this method finds images with the same ground truth class which the classifier scored the lowest of all such images, forming a low-confidence baseline.
It copies the background from one of these images instead of using a black/grey background
to try and put the masked image back into the distribution of images from the ground truth class.
This style of masking is used to compute correctness.

(approach - consistency)
An explanation is consistent if it is invariance w.r.t. a number of mostly semantically invariant transformations.
These include small affine transformations, horizontal flips, vertical flips, and adding noise.

(approach - confidence)
An explanation is confident if the masked images it produces still have high condidence under the classifier.
Masked images are produced as for correctness, by copying a distractor from the same class into the background.

(experiments)
The experiments compare existing explanations (LIME, Grad-CAM, Integrated Gradients, SmoothGrad) using the proposed metrics.
1. Correctness: Classifiers have higher accuracy on explanation-masked images than on images they were least confident on (the ones used to fill in the background).
2. Grad-CAM is most correct, followed by SmoothGrad, Integrated Gradients, and LIME.
3. Consistency: Grad-CAM explanations are most resilient to the proposed transformations with Integrated Gradients, SmoothGrad, and LIME being successively less invariant.
4. Confidence: Explanation-masked images have higher scores for their ground truth class than the low-confidence baseline images.
5. Hyperparameter variations in the correcness/confidence metrics mostly preserve the ranking of methods, though the absolute values of performance do change substantially.

(conclusion)
The paper concludes that Grad-CAM is usually the best of the methods tested according to the new metrics and that LIME is the worst.


Strengths
---

I really like the related work section. It could be a valuable resource going forward.

I like the research direction of this paper very much. I think that enumerating a suite of complementary benchmarks is a good way to measure explanation quality because we can only come up with benchmarks that capture a small part of what we want so far.


Weaknesses
---


I see some major conceptual flaws with these metrics:

* In section 3.1 it seems like the first reasons that normal masking failed is not solved by the proposed approach. The generated images are still out of distribution because the "foreground" and the "background" don't match.

* I'm concerned about the low-confidence distractor images used in the background. They are from the same ground truth class as the high confidence images they are pasted into the background of, correct? The correctness metric is supposed to capture whether or not an explanation highlights all the class-relevant content in an image and no more. However, information that the explanation did not highlight (the background) can inform the classifier of the ground truth class because the background came from an image of that class (even if a low confidence one). This is especially true because the relevant objects might be in differrent positions in the two images. Thus it could be that the explanation did not highlight informative content but the classifier still gets the corresponding masked image correct because of the background. How often does this happen?

* Consistency is supposed to measure "the ability of the explainer to capture the relevant components" under semantically invariant transformations.
The reported metric is mimized when the explanation is the same before and after a variety of transformations.
If this were the case then at least one of them must be wrong in the sense that it would not have captured some relevant components
(unless perhaps it just highlighted everything and was thus useless).
Because of the transformation (e.g. 15 degree rotation) the relevant components would have been at a different position, but the best explanation according
to this metric would have been at the same position. Thus this metric seems to reward explanations for not capturing relevant components.


Parts I Didn't Understand:

* In section 3.1, I don't understand the second reason that masking failed. In what sense is masking made meaningless? How is that sense different from the out of distribution concern from the first point?


Missing Details / Presentation Weaknesses:

* Missing reference to [1] which provides more metrics.

* The meaning of confidence is different than it normally is and this may be confusing.
Neural networks should be well calibrated, not necessarily confident (in the commonly used sense of [3]).


Minor flaws:

* Masking by replacing the background with grey (i.e., the bias of the first conv layer) rather than black is more common (e.g., [2] and Grad-CAM). A grey background negates the bias. It's not clear that the background should cancel the bias, but it would be nice to compare to both grey and black masking in Table 7.


[1]: Adebayo, Julius et al. “Sanity Checks for Saliency Maps.” NeurIPS (2018).
[2]: Zeiler, Matthew D. and Rob Fergus. “Visualizing and Understanding Convolutional Networks.” ECCV (2013).
[3]: Guo, Chuan et al. “On Calibration of Modern Neural Networks.” ICML (2017).


Final Evaluation
---

This paper relies solely on theoretical arguments to show its metrics capture meaningful information. Empirically, it only shows that the proposed metrics can differentiate between some popular explanations. It does not empirically show that the differentiation is meaningful (e.g., by measuring agreement with human judgement). This by itself isn't a problem. However, above I detailed significant flaws in the theoretical justification for the metrics, so I can't recommend these metrics (this paper) on either a theoretical or an empirical basis.

Quality: Per above, I do not think the arguments/evidence in the paper support its conclusions.
Clarity: The paper could be clearer, but can be understood without too much effort.
Originality: These metrics are new enough, being novel variations on prior approaches.
Significance: If I was convinced the metrics made sense then I would guess this paper would be very impactful. As is, I don't think it will have much impact.

The quality of the paper is my reason for the low rating. I'm interested to see whether what others think to make sure I've understood the paper correctly and analyzed it accurately. If my understanding is incorrect I could definitely raise my rating.

Post-Rebuttal Evaluation
---
After reading the other reviews and the author responses and taking a brief look at the updated paper I still think this paper should be rejected.

The authors' response to my comments clarified my understanding of the consistency metric. Now I understand it and think it is a useful metric.

However, I did not find clarification about the confidence or correctness metrics, though I agree they are not redundant. They still don't really quite make sense to me. This puts me in about the same position as R3, who also doubts those metrics. In the end, this leaves my initial evaluation essentially unchanged. I still recommend rejection because the paper relies on a theoretical understanding of what makes confidence and correctness metrics useful and that understanding is not provided.

**Experience Assessment:**

I have published one or two papers in this area.

**Review Assessment: Checking Correctness Of Derivations And Theory:**

I assessed the sensibility of the derivations and theory.

**Review Assessment: Checking Correctness Of Experiments:**

I assessed the sensibility of the experiments.

**Review Assessment: Thoroughness In Paper Reading:**

I read the paper at least twice and used my best judgement in assessing the paper.

---

> ### Author Response · Authors · 2019-11-15
> **Response to ReviewerII**
>
> Thank you for your kind words and the succinct summary  of our paper. We would like to address your comments one by one.
>
> Comment 1: Proposed masking generates image that are still out of the data distribution.
> Response:
> As we have reported in table 9, doing normal masking (with black/gray pixels) doesn’t give us any accuracy or confidence improvements. Thus it does not help us at comparing different explainability methods. Since the background of the masked image is taken from an image of the same class in our dataset, the latter is more relevant than a constant value background and helps us in preserving spatial correlations that CNNs rely on to make their predictions. Moreover, by pasting salient pixels from a high confidence images to a low confidence one, we can observe an accuracy/confidence increase which is directly correlated to the goodness of an explainer (i.e., the accuracy increase is due to the fact that the explainer captured relevant pixels on the high confidence image). We have further elaborated on this in the responses below.
>
> Comment 2: Concerns about information leakage from background image used in masking.
> Response:
> We would like to emphasize that we do not consider the explanation map of the low confidence image. The latter is only used as a background onto which pixels highlighted by the explainer on the high confidence image are pasted.
>
> Additionally, we would like to summarize our correctness evaluation method below to clarify any further confusions.
>
> Our hypothesis is that, if the explainers are indeed capturing only the relevant pixels and nothing more, we should see an increase in accuracy if we pass only the relevant pixels as the neural network will not be confused by non-relevant pixels. Additionally, we should see a very small change in accuracy if we pass ONLY the non-important pixels as there is little to no information in these pixels for the discriminative task.
>
> We further performed experiments to show that a simple masking with empty pixels as background was not sufficient to see the above expected phenomena as the resultant images are out of distribution as well as not locally correlated (i.e., nearby pixels did not tend to have similar pixel values).
>
> Thus we proposed a new masking technique whereby we use low-confidence (as measured by the network’s prediction) images as background for the high confidence images belonging to the same class.
>
> We first measure base-line accuracies for the set of low-confidence images (say BL).
>
> For the normal masking experiments, we take the relevant pixels from the high confidence images and paste them on top of the low confidence images and remeasure the accuracies (say I).
>
> Similarly, for the inverse masking experiments, we take the non-relevant pixels from the same high confidence images and paste them on top of the low confidence images and measure the accuracies again (say II).
>
> We define correctness as the changes in accuracies in the above two experiments (Table 2 in the paper) (i) between the baseline and masked images (ii) baseline and inverse masked images.
>
> Comment 3: Concerns about how consistency is evaluated
> Response:
> The reviewer points out that Consistency is minimized when “the explanation is the same before and after a variety of transformations.”. This seems to be an unfortunate misunderstanding. We have described in the first paragraph of Sec. 3.2.2 that a consistent explainer would produce similar explanations modulo transformations for the transformed images. Therefore, we perform an inverse transformation (t^{-1}) before we compute the difference between the generated heatmaps. We have mentioned this in Eq. 3 as well. In this case we only reward the explainer when the explanation that it yields is still consistent under different semantically invariant transformations.
>
> For example, if we apply vertical flip as a semantically invariant transformation, we flip the generated heatmap from the transformed image before comparing with the heatmap generated by the untransformed image.
>
> Comment 4: Effect of grey masking instead of black masking
> Response:
> We have added experimental results for this configuration in the appendix. You can find them in table 10. We find that using the grey background does not solve the issue either.
>
>
> We hope these explanations address your concerns satisfactorily.
>
> Regards,
> The Authors

---

### Official Review · AnonReviewer1 · 2019-10-24
**Official Blind Review #1**

**Rating:** 3

**Review:**

--------- AFTER rebuttal

1) "We identify issues with current masking procedures as proposed in other papers"

One of the major issues with the current masking procedure is that the resulting image is out of the data distribution. Even though your method achieved high accuracy in Table 2 for correctness, the generates images is still out of the data distribution.

2) "We propose a cost-effective masking technique that doesn’t require retraining of the underlying classifier"

The authors compared against zero and gray masking for correctness. None of those masking methods require retraining of the underlying classifier. It is not clear, which previous masking technique required retraining of the underlying classifier?

3) We also further show that when performing a comprehensive evaluation, there is no one clearly better explainer and thus practitioners need to be careful about which explainer they choose.

This is an observation made upon through exploratory analysis and is not a technical novelty.

4) " Confidence on the other hand, also informs us about the per-instance behavior."

The confidence measures the change in probability assigned to the ground truth class. Table 3 should also show the variance in the confidence to understand the instance-level behavior.

The experiments given in the paper, it looks like confidence and correctness are positively correlated. An example of the model where they are not positively correlated will help the reader understand the importance of each of these terms.

5) " Effect of Thresholding on results"
Thank you for the explanation and new experimental results.




------------------------- BEFORE rebuttal
The paper proposed different metrics for comparing explainers based on their correctness (ability to find most relevant features in an input, used in prediction), consistency (ability to capture the relevant components while input is transformed), and the confidence of the generated explanations. To evaluate correctness, the authors proposed to study the change in the classification accuracy of the target model, under a perturbed dataset where the most relevant regions (as given by explainer) of the image is preserved and the remaining content is replaced with non-informative backgrounds for the target class. For consistency evaluation, the authors proposed to apply transformations like rotation, translation and flip that doesn’t semantically change the input image. For confidence evaluation, they compared the prediction performance on the original image, masked image (only salient regions) and inverted masked image (only non-salient regions).

Major
•	The paper lack technical novelty.
•	The confidence component looks redundant and can be incorporated in the correctness component.
•	The inverse saliency map idea is already proposed in “Evaluating the visualization of what a deep neural network has learned” for evaluating saliency maps. There the authors gradually replace the most salient regions with random noise and observe a decrease in prediction accuracy.
•	Most of the saliency maps producing methods, generate continuous maps. For making, we need to convert the continuous map to binary by using a threshold. An analysis of choosing different values as threshold is missing. By choosing an appropriate threshold, the size of the most salient region can be controlled. Thus, although Grad-Cam spread saliency over large area, we can use a higher threshold to define the binary mask.
•	Grad-CAM, integrated grad and smooth grad are all gradient-based saliency maps. There are perturbation-based saliency maps, which aims to find most salient regions such that removing those regions produce a maximum drop in prediction accuracy. Example “Interpretable Explanations of Black Boxes by Meaningful Perturbation.”, “Object detectors emerge in deep scene cnns” .  An evaluation of such methods is missing.

Minor:
•	The text in the figures has very small font size and is not readable.
•


**Experience Assessment:**

I have read many papers in this area.

**Review Assessment: Checking Correctness Of Derivations And Theory:**

N/A

**Review Assessment: Checking Correctness Of Experiments:**

I assessed the sensibility of the experiments.

**Review Assessment: Thoroughness In Paper Reading:**

I read the paper thoroughly.

---

> ### Author Response · Authors · 2019-11-15
> **Response to Reviewer1**
>
> We would like to thank the reviewer for their valuable insights. Please find below our responses addressing your concerns.
>
> Comment 1. Lacks Technical Novelty
> Response:
> We would like to emphasize our major contributions:
> 1. We identify issues with current masking procedures as proposed in other papers
> 2. We propose a cost-effective masking technique that doesn’t require retraining of the underlying classifier
> 3. We identify the one-dimensionality of current research into evaluating explainers, proposing other important components.
> We also further show that when performing a comprehensive evaluation, there is no one clearly better explainer and thus practitioners need to be careful about which explainer they choose.
>
> Comment 2: Confidence is redundant relative to correctness.
> Response:
> While the two numbers do come from the same experiment, we would like to point out that they are complementary to each other. Correctness gives us a coarse, distribution level view, i.e, how well the explainer performs on average. Confidence on the other hand, also informs us about the per-instance behaviour. In some instances, especially in the medical diagnosis domains, per-instance behaviour is more important than a distribution-level statistic. Additionally,  if method A and B have similar correctness scores, Confidence and Entropy (reported in table. 3) gives us more in-depth information about the per-sample performance of these methods, allowing us to differentiate between the performance of both models
>
> Comment 3: Inverse Saliency is already proposed
> Response:
> We are aware of the paper referred to by the reviewer. In fact, we have cited it in the related work section as well. However, our approach differs from it in the following ways.
> 1. We do not add noise when computing the inverse saliency maps
> 2. We do not perform an iterative process to compute AuC, instead we compute the inverse only once. In the experiments, we repeat the matching process multiple times only to remove any potential correlations between the randomly chosen background and foreground images. This is an additional step and is not integral to our proposed method.
> 3. Unlike the referred paper, we perform the inverse masking to make sure that the explainers have not missed any relevant pixels and is only a small part of our proposed framework.
>
>
> Comment 4: Effect of Thresholding on results
> Response:
> The reviewer is right in pointing out that the size of the generated explanation depends on the thresholding used. We conducted experiments using different percentile-based thresholds and found that the relative trends between the explainers does not change with the threshold. We have included the results in Table 6 and 7 in the Appendix
>
> Comment 5: Evaluating Perturbation based methods
> Response:
> In addition to gradient based methods like GradCAM, SmoothGrad and Integrated Gradients, we also evaluated our metric suite on LIME, the most widely used perturbation based explanation method for our comparison. The other perturbation methods can be similarly evaluated as well.
>
> We would like to emphasize that our work focuses on proposing a comprehensive suite of metrics to evaluate explainers and performing a comparative study using common explainers to show the suite’s behaviour. An exhaustive comparison of every explainer is unfortunately out of the scope of this paper but we do consider it as an important future work that can be collaboratively taken up by the XAI community.
>
>
> Thanks
> The Authors

---

### Public Comment · ~TING_TING_SUN1 · 2019-10-09
**Some questions**

Hi! Nice work on evaluation. I have a few questions.

In section 3.2.1, you use a special method to generate a masked dataset. You claim that these masked images belong to the data distribution. But in my opinion, some masked images of Figure 1 are still unnatural.  Is it possible that these masked images are also out of the data distribution, just like the images with empty pixels?

In the experiments of evaluating consistency, you use rotations as semantically invariant transforms. But neural networks are not invariant to rotations. Is this an evaluation towards explainability algorithms on rotation robustness?

---

> ### Author Response · Authors · 2019-10-24
> **Clarifications**
>
> Dear TING TING SUN,
> Thank you for your kind words and your comments.
>
> You are right in pointing out that the masked images could still be out of distribution. We hypothesized (in Sec. 3.2.1) that our masking provides samples that are closer to the data distribution than those with a black / blank background.
>
> We have further calculated the Inception Score[1] and FID[2] of the samples produced by our method and the black background. These results will be included in an update during the rebuttal phase.
>
> Here's the table of the scores we computed.
>
> Inception Score
> Explainer	       Black background  Our Method
> Integ. Grad   	        21.01		  89.1564
> SmoothGrad	  	24.4578	          137.6948
> GradCAM 		231.4284		  428.9047
> LIME 			60.2835		 137.9088
>
>
> FID Score
> Explainer	       Black background  Our Method
> Integ. Grad 	  108.8858	 	66.0593
> SmoothGrad 	   91.0726		46.9035
> GradCAM           409.6676		1.0368
> LIME		   62.2233		32.9805
>
>
> As you can see, in both the metric, our method outperforms the simple blank pixel background.We see from these results that, indeed our method produces inputs closer to the data distribution.
>
> Regarding your note on using rotations as a sematically invariant transform while evaluating consistency, our decision to include the rotation was motivated by the fact that real-world objects under rotation remain unchanged to an human observer (expect for rare cases like the digit 9). This decision was not influenced by whether neural networks were invariant to the transformation or not.
>
> Furthermore, we only take those inputs for which the prediction of the underlying classifier does not change under the transformation as described in Sec. 3.2.2. Additionally, we also showed that the classifier accuracy on the entire dataset doesn't change significantly under the rotations we consider (Please see Table. 1 in section 4).
>
> Finally, we would like to re-emphasize that this is an evaluation towards the explainability algorithms. In fact, we keep the underlying classifier unchanged under all the experimental setups. This allows us to factor out the any influence that pathologies of the underlying classifier might have on the observed results.
>
> Regards,
> The Authors
>
> [1] Improved Techniques for Training GANs https://arxiv.org/abs/1606.03498
> [2] GANs Trained by a Two Time-Scale Update Rule Converge to a Local Nash Equilibrium https://arxiv.org/abs/1706.08500

---

### Author Response · Authors · 2019-11-15
**Summary of changes to the paper**

We would like to thank the reviewers for their valuable feedback.
The following are the changes to the paper that we have made in response to their comments.

1. Added experimental results on the effect of thresholding on the masking process.
2. Added experimental results on the effect of grey background as compared to our proposed masking technique.
3. Added additional equations to clarify how our metrics are computed.
4. Added more more images in the appendix that exemplify our proposed masking used in the correctness metric.
5. A clearer explanation of how the consistency metric is evaluated.
6. More in-depth discussion into why confidence and correctness are not redundant.
7. Minor rewrites to fix grammar and typos.
8. Added additional citations to related works.

We hope that these address the reviewers' concerns and that they would reconsider their evaluation of our work.

Regards,
The Authors

---

### Decision · Program_Chairs · 2019-12-19

**Decision:**

Reject

**Comment:**

The paper proposes metrics for comparing explainability metrics.

Both reviewers and authors have engaged in a thorough discussion of the paper and feedback. The reviewers, although appreciating aspects of the paper, all see major issues with the paper.

All reviewers recommend reject.